# ACFun: Abstract-Concrete Fusion Facial Stylization

**Jiapeng Ji, Kun Wei,*Ziqi Zhang, Cheng Deng**
School of Electronic Engineering, Xidian University
Xi'an 710071, China
`jiapengji777@gmail.com, weikunsk@gmail.com, zqzh9116@gmail.com,`
`chdeng.xd@gmail.com`

## Abstract

Owing to advancements in image synthesis techniques, stylization methodologies for large models have garnered remarkable outcomes. However, when it comes to processing facial images, the outcomes frequently fall short of expectations. Facial stylization is predominantly challenged by two significant hurdles. Firstly, obtaining a large dataset of high-quality stylized images is difficult. The scarcity and diversity of artistic styles make it impractical to compile comprehensive datasets for each style. Secondly, while many methods can transfer colors and strokes from style images, these elements alone cannot fully capture a specific style, which encompasses both concrete and abstract visual elements. Additionally, facial stylization often alters the visual features of the face, making it challenging to balance these changes with the need to retain facial information. To address these issues, we propose a novel method called *ACFun*, which uses only one style image and one facial image for facial stylization. *ACFun* comprises an *Abstract Fusion Module (AFun)* and a *Concrete Fusion Module (CFun)*, which separately learn the abstract and concrete features of the style and face. We also design a *Face and Style Imagery Alignment Loss* to align the style image with the face image in the latent space. Finally, we generate styled facial images from noise directly to complete the facial stylization task. Experiments show that our method outperforms others in facial stylization, producing highly artistic and visually pleasing results.

## 1   Introduction

Style transfer is a long-standing research topic that aims to generate a new artistic image from an arbitrary input pair of natural images and painting images by combining the content of the natural image and the style of the painting image. With the rapid development of deep learning, optimization-based methods [1–3] and feed-forward based methods [4, 2, 5, 6] continually emerged. These methods have made progress in visual quality and computational efficiency. However, these style transfer techniques can only modify low-level image features such as strokes and colors, and they are unable to adjust high-level semantics such as shape and content. It's like using the style image as a template to paint the content image instead of truly creating a new image based on the style. Style encompasses not only basic visual elements like strokes and colors but also deeper aspects such as structure, layout, and shape. These elements combine to create a unique visual language and form of expression. In the context of facial stylization, the challenges of style transfer are even more significant. Facial recognition relies on distinct and recognizable individual characteristics, which means that style transfer must capture the visual features of the style image and seamlessly integrate them into the target image while preserving facial recognizability.

With the rapid development of diffusion model-based image generation methods, several style transfer methods [7–9] have emerged, but these methods usually require a set of images belonging to the same

---

*Corresponding author.

| Style Image | Face Image | Result | Style Image | Face Image | Result |
|:---:|:---:|:---:|:---:|:---:|:---:|

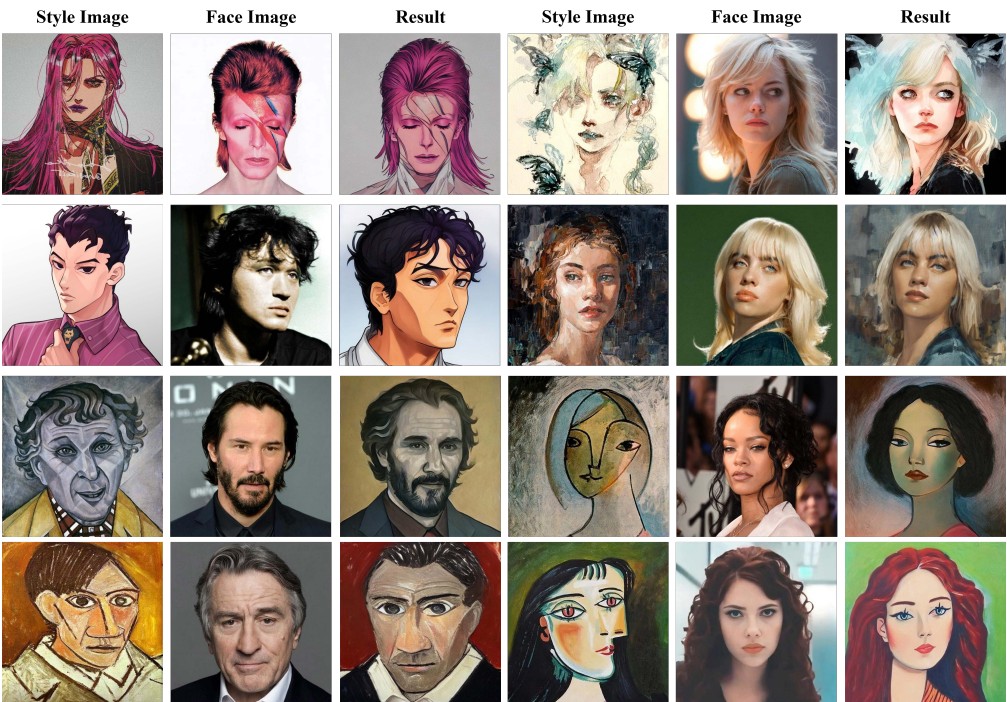

Figure 1: As shown in the figure, the ideal result of facial stylization aims to ensure the faithful restoration of facial information while also significantly deforming the facial image according to the given style image. To achieve high-quality results, it is essential to ensure that the production process not only meets the standards for good generation quality but also encompasses a compelling and evocative artistic atmosphere.

style for training. However, due to the scarcity of high-quality stylized images, it is difficult to obtain them. Moreover, the diversity and complexity of artistic styles make compiling a comprehensive dataset for each style impractical. Subsequently, there are also some Textual-Inversion methods [10, 11] that can use a single image, but this method only fine-tuned the CLIP model as the conditional input, which limited its ability to generate models and often caused it to be heavily affected by style degradation, resulting in simpler results and reduced image quality. Moreover, its style transfer method of adding noise and denoising to the target image reduces the flexibility of the generated results and further limits its final generation performance.

As depicted in Figure 1, the most prominent challenge in facial stylization tasks is that images with artistic styles often display significant differences from real-world facial photos. These differences are evident not only in visual features but also in high-level semantics, such as shape and distribution. In particular, as shown in the lower part of Figure 1, there are notable disparities between abstract style portraits and real portraits, leading to substantial changes in facial features during the stylization process. On the other side of the coin, it is essential for stylized facial images to maintain consistency in facial information so that they remain recognizable to people. In summary, for stylized facial images, balancing the ability to maintain facial images with the drastic semantic changes brought about by stylization is the key to achieving good facial stylization results.

To this end, we propose a facial stylization method that allows deep neural networks to fuse abstract visual elements (deformation, distribution, atmosphere, etc.) from style and facial image with concrete visual elements (color, brushstrokes, lines, etc.) into an imagery visual feature just like artists, to achieve facial image stylization, rather than simply transferring texture and color information. Specifically, we introduced an Abstract-Concrete Fusion Facial Stylization Model (ACFun) to fuse, consisting of an Abstract Fusion Module(AFun) and a Concrete Fusion Module (CFun). The method ACFun uses for facial stylization involves learning abstract and concrete features separately and then combining style images with facial images by aligning them in an imagery feature space formed by the fusion of abstract and concrete features. We first extract abstract features of style and face

through AFun by encoding style images into the CLIP latent space and combining it with the facial description text prompts we use; then, we optimize it using a Textual-Inversion-based method to make it recognizable by CLIP for subsequent generation processes. In addition, by introducing face images during the training process, we implicitly fuse facial information with style abstract information. However, CLIP has limitations in distinguishing between specific style and facial images at the fine-grained level. To address this, we adopt the CFun module, which is a set of trainable parameters embedded in a pre-training generator to enhance its ability to extract concrete features from specific styles and facial images, which greatly improves the quality of the generated results. Finally, the above process is constrained by the Face and Style Imagery Alignment loss we proposed. We demonstrate the effectiveness and excellent facial stylization effect of our method through massive experiments.

## 2   Related Works

**Text-to-Image Generation.** The method based on generating adversarial networks [12] has been widely applied in text-to-image generation by training on paired images and text samples. [13] improves condition generation[14] by changing the input conditions from a single class to more complex text descriptions. However, due to the difficulty of training GAN models and the significant progress of large language models [15, 16] and diffusion models, the diffusion models have gradually replaced the position of GAN. The diffusion model[17, 18]can generate realistic images through multiple rounds of iterative refinement. It has been widely used in the image generation task and has achieved significant results [19], attracting the attention of many researchers. Among all these diffusion-based methods, Stable Diffusion [20] is the most typical work. It achieves awe-inspiring results through training on large-scale text image datasets. Afterward, SDXL [21] was introduced with larger model parameters and larger datasets. However, these methods all focus on learning text prompts, meaning that the model can only fit the given reference style through existing knowledge rather than train the generated model to have the ability to generate specific style images. We fine-tune the model to make it more intuitive to learn the target style straightly. Thanks to these powerful pre-trained generative models, we can flexibly apply them to various downstream task applications.

**Personalization of Generative Models.** How to use a pre-trained text-to-image large model to complete personalized image synthesis tasks has attracted widespread attention [22]. The widely used method in image generation tasks is the inversion-based method, which aims to find a corresponding noise or conditioning embedding to a generated image. Textual Inversion [9] and Hard prompt made easy (PEZ) [23] convert a set of images of an object into corresponding textual representations (such as embedding, token) instead of changing the parameters of the text-to-image model. Dreambooth [7] learns target concepts by fine-tuning the pre-trained model. At the same time, when learning new class concepts, Dreambooth transforms this concept into a special sub-concept of a specific class. Dreambooth achieves the goal of data augmentation by generating original classes, preventing new concept classes from overwriting existing original classes, leading to catastrophic forgetting and affecting the diversity of generated models, leading to overfitting. Lora [8] has also achieved widespread application by transferring the strategy of fine-tuning pre-trained LLM to image generation methods[24], allowing it to save computational and storage costs without the need to train the entire generation network. Although the above methods have achieved good results, they all learn the description of specific concepts through a set of images, which makes it difficult to meet our needs. The method we adopt only requires a pair of images to learn the abstract and concrete features of style images and facial images and ultimately generate a stylized facial image.

**Facial Stylization.** Facial stylization is currently a widely recognized task in the field of style conversion. Typical style transfer methods such as [25, 26, 4, 6, 27] can effectively transfer the concrete visual features such as texture and color contained in the given style image. However, when facing facial images, especially when transferring real facial images to styles with strong geometric distortions (such as comics, anime, abstract paintings, etc.), the effect is often poor. Although some methods [28, 29] use large-scale paired image data for image-to-image translation training, in practical applications, due to the diversity and scarcity of high-quality style images, collecting such a large number of paired images is not realistic, and some images do not even exist, which makes the above methods difficult to apply in practice. StyleGAN [30] achieves facial stylization through image inversion [31, 29, 32–34], but due to the presence of its latent space, it suffers from problems such as image degradation and limited deformation. Although the method of textual inversion [10, 11] can form certain deformations, the problem of style degradation cannot be ignored. Our method solves the above problem by learning abstract and concrete features separately and aligning images.

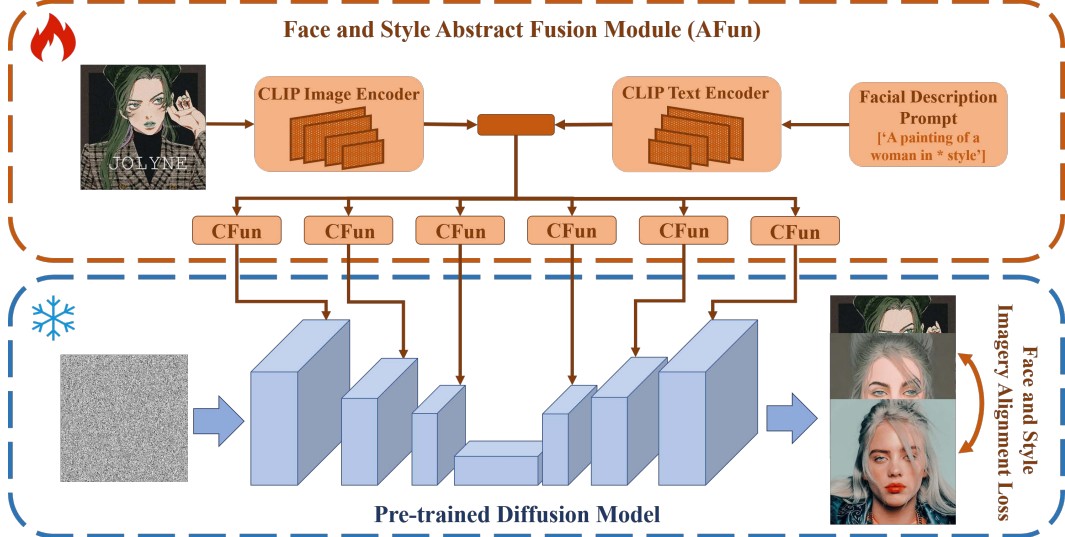

Figure 2: The overall schematic diagram of ACFun consists of two parts: one is the AFun module for extracting abstract features, and the other is the CFun module for extracting concrete features. The entire training process is controlled by our carefully designed Face and Style Imagery Alignment loss. We utilized a facial description text prompt to enhance the fusion of style and facial images.

## 3 Method

### 3.1 Overview

In this paper, we propose a novel method named ACFun for image style transfer; the structure and process are shown in Figure 2. Facial style transfer aims to preserve facial identity information while being as close to the target style as possible. Inspired by different artistic styles of expression, we divide visual features into more complex and difficult to describe abstract features, such as shape or atmosphere, and more specific concrete features, such as color or strokes. And ultimately, by combining abstract features and concrete features into imagery features and utilizing them to generate stylized facial images.

Therefore, we first propose an AFun module. By using a CLIP encoder to encode the style image and combine it with a facial description text prompt using the attention mechanism to fully use the information of the reference style image. Then, we optimize it into an embedding $e*$ that can be recognized by the generated model in the subsequent generation process. However, simply using the aforementioned Inversion-based method often leads to style degradation, resulting in the loss of some concrete visual content in the final result, which is often fatal for facial stylization tasks. So, both facial and style images are introduced simultaneously during the training process. To this end, we adopt the CFun module, to improve the flexibility of the network and enable it to have contract generation capabilities, we insert a set of trainable parameters into a pre-trained U-net and adopt the Face and Style Imagery Alignment loss to constrain the above process.

### 3.2 Face and Style Abstract Fusion Module (AFun)

To fully utilize the pre-trained large-scale model, we hope to learn the style in the target style image as a specific intermediate representation, which is suitable for the pre-trained model. This method uses CLIP as a text encoder for text-to-image generation in stable diffusion. It inputs the text embedding, which is embedded by the CLIP text encoder $\tau_t$ as a condition. We use the pseudo-word '$S^*$' as a placeholder for the image style and introduce the concept of specific image styles into the pre-trained model through a learnable corresponding embedding $e_t^*$. However, using only a single pseudo-word $S^*$ as a placeholder, this simple and vague text prompt inevitably leads to us only being able to extract the content of the entire style image, regardless of whether it is an abstract style feature or a specific content feature. This will bring problems due to the content of the style image in our subsequent

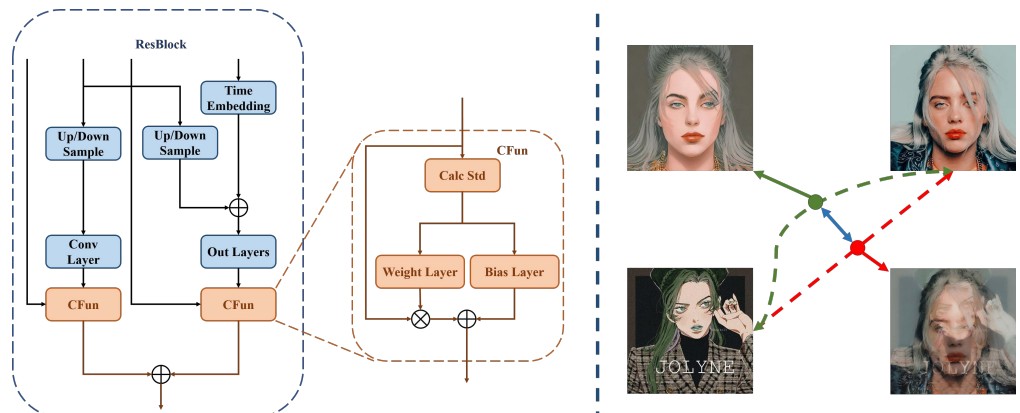

Figure 3: **Left:** The specific insertion method and structure of our CFun. **Right:** Our proposed imagery latent space, where the **red** line represents the VQ visual space, the **green** line represents the imagery latent space, and the **blue** line represents the process of mapping the original image from the VQ space into the imagery latent space through the introduction of abstract features $e_a$.

facial stylization process, such as style leakage or style coverage. To address this issue, we used facial description text prompts $S_f^*$. The style and content of the style image can be automatically separated by the description of the facial image, such as "A painting of a woman in $S^*$ style." This is because in the subsequent training process, style images $I_s$ and facial images $I_f$ are introduced, and CLIP naturally implicitly receives information from facial images $I_f$ during the feature update process. We embed the style image $I_s$ using CLIP image encoder $\tau_i$ as embedding $e_{i_s}^*$. Then, we fuse it with facial description text prompts embedding $e_{t_f}^*$ to obtain abstract features $e_a$. This process can be expressed as:

$$\min_{e_{t_f}=e_a} E_{I_s,I_f} E_{z_t \sim q(z_t|I_s,I_f)} \|\epsilon - \hat{\epsilon}_\theta(z_t, t, \tau_t(y, S_f^*))\|_2^2, \tag{1}$$

where $\epsilon$ stands for an initial noise map $\epsilon \sim N(0,1)$, $\hat{\epsilon}_\theta$ is the denoising network, $t = 1 \dots T$ stands for the time step for the diffusion model. Since our method only requires a pair of images to complete training, it is difficult to thoroughly learn the style in the background image using a vanilla textual inversion method, and overfitting is prone to occur. To alleviate this problem, we also introduced image information, allowing the model to learn the style and facial abstract visual elements in the image more fully. Benefiting from the CLIP model that has already aligned text with images in its latent space as a multi-modal model, the fusion process of style image embedding $e_{i_s}^*$ with facial description text prompts embedding $e_{t_f}^*$ using an attention mechanism can be represented as:

$$Q_n = W_Q^{(n)} \cdot e_{t_f}^{(n)}, K = W_K^{(n)} \cdot e_{i_s}, V = W_V^{(n)} \cdot e_{i_s},$$
$$e_a^{(n+1)} = Attention(Q_n, K, V) = softmax\left(\frac{QK^T}{\sqrt{d}}\right) \cdot V, \tag{2}$$

and set dropout in the attention mechanism to alleviate overfitting problems.

### 3.3 Face and Style Concrete Fusion Module (CFun)

Even if we have obtained an abstract feature $e_a$ that blends style and facial information, relying solely on this feature to generate images means that we can only fit the target style and facial image using the prior knowledge already present in the pre-trained model. However, in real life, there are numerous style images and facial images, many unseen by pre-trained models, making it challenging to generate the desired results. To solve the above problem, we propose the CFun module for generating concrete visual features. By inserting a set of learnable parameters into U-net, we enhance the flexibility of the network and enable it to learn the visual features of input images. For its insertion position, considering that we do not want to damage the style and information in the facial image excessively. It is too risky to directly insert parameters into the Attention Block for training, as this

may involve overly advanced semantics. Directly changing the Attention Block may result in too drastic advanced semantic changes, ultimately leading to excessively distorted stylized face images. So, we choose to insert the parameters into Resblock, which can make our parameters focus more on specific concrete visual features and indirectly affect the Attention module, thus achieving a balance between the information retention ability of facial images and the high-level semantic changeability brought by stylization. In addition, inspired by AdaIN [4], our insertion module adopts an affine mechanism, which can ensure that the network parameters are not affected during initial training, further improving the stability during training. The specific structure and insertion method are shown on the left of Figure 3. We separate content from the style by calculating mean-standard:

$$CFun(x) = \gamma(\sigma(x))x \oplus \beta(\sigma(x)). \tag{3}$$

And only the style information, i.e., standard $\sigma(x)$, is fused with the network. This is mainly because the abstract features we use are only fused from style images and facial description text prompts at the beginning of training, and no facial image information has been obtained yet. To avoid excessive activation of abstract features and style images in cross-attention during the training process, causing the generated stylized facial images to irreversibly guide to style images, resulting in the final generation result being strongly covered by style.

### 3.4 Face and Style Imagery Alignment Loss

Finally, we use a Face and Style Imagery Alignment loss to constrain all the above processes. However, as shown on the right of Figure 3, even if encoding using VQ space, the interpolated images represented by simple style images and facial images are meaningless, just a simple superposition of the two images. However, initially, we had already used AFun to fuse style images with facial description text prompts and provided an abstract feature $e_a$ for subsequent training processes. This allowed us to have abstract feature $e_a$ guidance in the subsequent optimization process. After combining the abstract feature with initial noise, the space it projects onto is not simply a VQ latent space composed of simple image features but a latent space composed of semantic features after the combination, we call this latent space, which combines abstraction and concreteness, the imagery space. This also makes our proposed Face and Style Imagery Alignment loss effective. Specifically, by aligning $I_s$ and $I_f$ in the imagery space, we obtain an image $I_i$ that integrates abstract and concrete features of facial image $I_f$ and style image $I_s$ to achieve the goal of facial stylization. We also use the two hyperparameters we establish, $\beta$ and $\gamma$, to control the tendency of the model towards the relationship between facial image content and style during the optimization process. Finally, the pre-trained text-to-image diffusion model $DM$ is used, and the initial noise map $\epsilon \sim N(0, 1)$ and the abstract feature $e_a^*$ obtained in the previous step are given as inputs to generate the image $\hat{I}_i = DM(\epsilon, e_a^*)$, and the diffusion model is fine-tuned using the squared error loss. The specific formula is as follows:

$$\min_{\hat{I}_i} E_{I_i, \epsilon, t, e_a^*} [\beta \omega_t \| DM(\alpha_t I_i + \sigma_t \epsilon, e_a^*) - I_s \|_2^2 + \gamma \omega_t \| DM(\alpha_t I_i + \sigma_t \epsilon, e_a^*) - I_f \|_2^2]. \tag{4}$$

Where $\alpha_t, \sigma_t, \omega_t$ are parameters used to control the noise schedule and sampling quality and are functions of the diffusion process time $t \sim U([0, 1])$. Meanwhile, this is a win-win training strategy. On the one hand, the abstract features $e_a$ of AFun provide a good latent space for the training of the generator. On the other hand, through gradient backward, the abstract features of AFun can also extract concrete information from facial images, providing better quality guidance. Through this method, we can constrain the optimization process of abstract feature fusion for style images and face images in CLIP space, as well as the extraction of concrete features for style images and face images using learnable parameters inserted into the generator.

## 4 Experiments

In this section, we conduct experiments using images collected from the Internet and provide visual comparisons. We also conducted experiments on both facial stylization and text-to-image generation tasks. Our method has achieved good visual effects in various experiments, demonstrating its effectiveness.

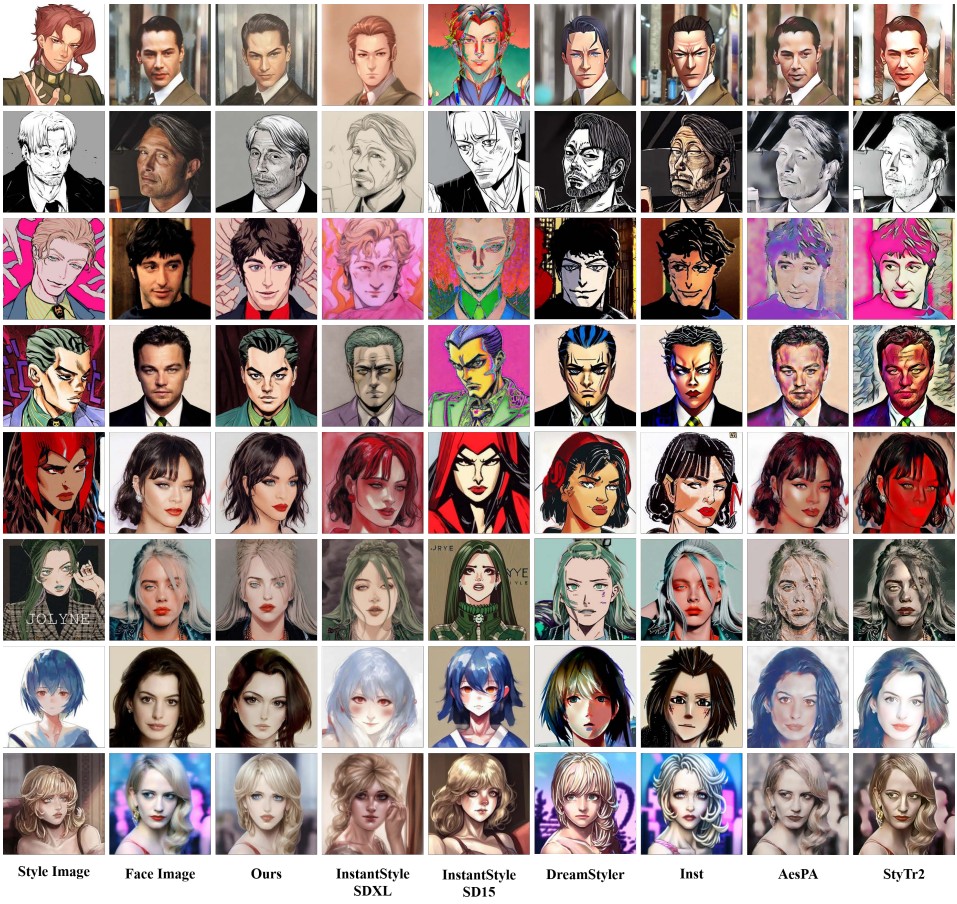

| Style Image | Face Image | Ours | InstantStyle SDXL | InstantStyle SD15 | DreamStyler | Inst | AesPA | StyTr2 |

Figure 4: We experimented with different facial and stylistic images and compared our previous high-performance methods. It can be seen that our facial stylization results have a stronger style, while ensuring facial information while harmoniously and naturally integrating with the target style.

## 4.1 Implementation Details

We trained on a single Nvidia A6000 graphics card, and in the case of a single pair of images, we set the batch size to 1. Each epoch took about 30 seconds, and the overall training process is 5 epochs, which takes only 3 minutes. We set the base learning rate to $1.0e - 04$, and the remaining hyperparameters are consistent with Stable Diffusion without changing. Through 40 steps of diffusion, our method can obtain stylized facial images with good results. We set the hyperparameters $\gamma$ and $\beta$ to 0.8 and 1.0, respectively, and all subsequent experiments will use this hyperparameter setting method.

## 4.2 Quality Analysis

We compare our method with the state-of-the-arts image style transfer methods, including InstantStyle [35], DreamStyler [10], Inst [11], AesPA [36], StyTr2 [37]. As shown in Figure 4, the generation results fully demonstrate the excellent performance of our method. Compared with the traditional two-column method on the far right, it can be seen that our method can modify the target image according to the high-level visual elements of the reference style image instead of traditional methods only focusing on low-level visual elements such as color or strokes. Compared to the DreamStyler and Inst inversion-based methods, it can be observed that both methods suffer from serious style degradation, which makes the stylized facial images they generate more sloppy and tend to generate simpler images. InstantStyle pursues fast generation due to its Tuning-Free mode. However, due to its use of ControlNets [38] to control the final generated face, the model only pieces together the

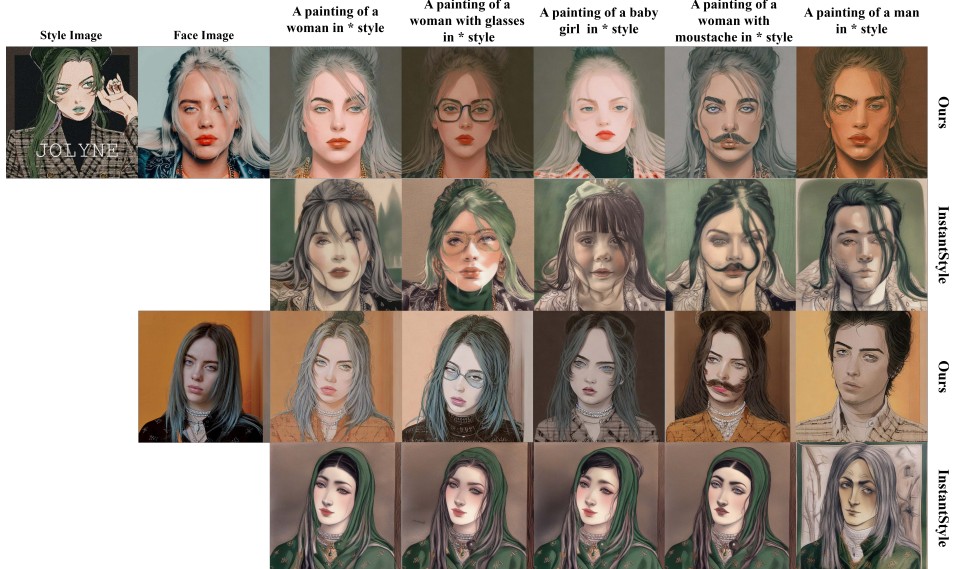

Figure 5: We have made more comparisons with the InstantStyle methods. Our method not only achieves good results in terms of generation quality but also has the ability to generate following text guidance.

elements in the style image according to the guidance by ControlNets rather than truly understanding the face. At the same time, due to the lack of new knowledge introduction, the generated results are not ideal. When using SDXL [21], the generated results also show some competitiveness. However, when using a smaller network like SD1.5, it can be seen that due to the insufficient prior knowledge to support its generation of input images of any style, the results are quite struggling and even generate a lot of noise in some images. On the contrary, although our method only uses SD1.4, a more basic pre-trained generative network, as our backbone model, the injection of new knowledge makes our generation results more effortless. It can be seen that our results faithfully restore facial image information while being close to the reference style image, displaying a highly artistic facial stylization result. We show more quality results and quantitative results in the supplement material.

### 4.3 Text-to-Image Generation

In addition to focusing on facial stylization generation, our method also has the ability to generate by text guidance since the Stable Diffusion we take as our backbone is originally a text-to-image generation mode. We compare our method with InstantStyle [35] as shown in Figure 5. It can be seen that, our method achieves better visual effects compared to Instantstyle. While ensuring that the style and facial features remain unchanged, we generated corresponding results based on the given text input and displayed good image quality. As Instantstyle is a method of tuning for free, it only relies on changing the attention parameter and subsequently using controlnet to control the final generation result. This also means that the Instantstyle method does not require additional knowledge injection, which makes it lack understanding of the given style and face. On the other hand, the presence of ControlNets can also interfere with the generation results of text participation guidance to some extent. As shown in the bottom line, Instantstyle hardly responds to the first four text inputs. Thanks to the injection of additional knowledge and the high compatibility between directly generating images from noise and text-to-image generation, our method can generate high-quality images that combine style, facial information, and fit text descriptions.

### 4.4 Ablation Study

We conducted ablation experiments on different modules to demonstrate how we extract the required features and how they work. As shown in Figure 6, we first do not use any CFun modules but only use abstract features from Afun for generation. It can be seen that the generated result is only a blurry

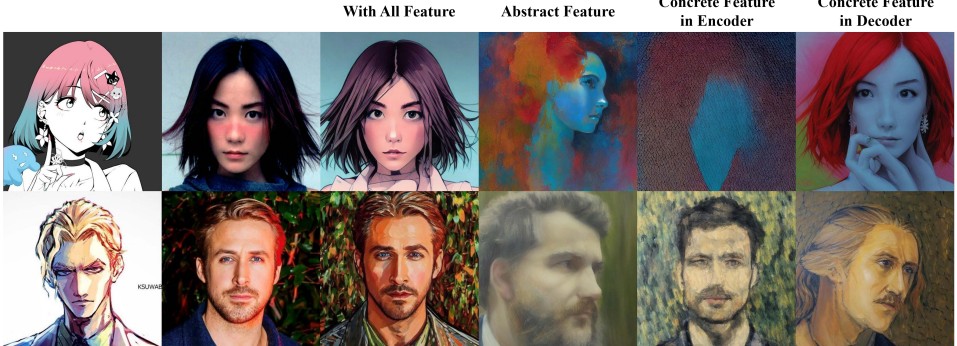

Figure 6: The ablation experiment demonstrated the role of our proposed abstract and concrete features, demonstrating the effectiveness of our proposed separation of learning abstract and concrete features.

image without specific facial information. However, at an abstract level, it can be seen whether it is male or female, and it also provides the colors required for the final generated image. By introducing Cfun into the encoder, it can be seen that the specific features of CFun provide low-level visual information, such as stroke texture. In the decoder, CFun can see that the main concrete features are the structure and posture of the face. The results of the ablation experiment indicate that the feature extraction mechanism is consistent with what we described in the methods section, proving the effectiveness of our method and the rationality of the proposed method. We show more results in our supplement material.

## 5 Conclusion

We propose a new facial stylization method called ACFun, which can achieve facial image stylization using only one image pair. To adapt to the task of facial stylization, we separate the learning of abstract features and concrete features through AFun and CFun and constrain the above process through a carefully designed Face and Style Imagery Alignment loss. We can transform any facial image into a specific style through short-term training. Numerous experiments have shown that our method exhibits excellent facial stylization and text-guided image generation results compared to state-of-the-art methods. A large number of visual results demonstrate that our method not only has good facial preservation ability but also generates stylized facial images with a strong artistic atmosphere. Meanwhile, the results of ablation research have demonstrated the effectiveness of our proposed method and demonstrated that our proposed AFun module and CFun module indeed learn abstract and concrete features separately. Our method provides excellent convenience for facial stylization generation.

## 6 Limitation

Even if we use separate learning methods, style is always difficult to describe, which often leads to some low-level or advanced visual features being deeply bound to certain content, resulting in inevitable style leakage problems. A typical example is that stylized facial images will change with the posture in the style image. Furthermore, due to the ambiguity of the style, our method is still unable to accurately extract a few visual elements of the style we want to exist in the style image. Moreover, our method relies on the effectiveness of the pre-trained model. We found that the training data used in the pre-trained model is imbalanced, especially when facing certain specific character images or style images. This can lead to serious semantic binding phenomena, making it difficult to stylize or causing obvious style leakage, which can also impact the performance of our method.

## 7 Acknowledgements

Our work is supported in part by the National Key R&D Program of China (No. 2023YFC3305600), Joint Fund of Ministry of Education of China (8091B022149), National Natural Science Foundation of China (62132016, 62406238), and Natural Science Basic Research Program of Shaanxi (2020JC-23).

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

# A    Quantitative Analysis

We also conducted quantitative experiments based on user study, shown in Table 1, providing 40 pairs of style facial images and their generated results, and collected voting results from 50 volunteers. The voting was conducted from three aspects: style consistency, facial consistency, and overall which image users prefer. The results are shown in the table. It can be seen that our method can achieve good results in style consistency and outstanding performance in facial consistency and overall evaluation. In addition, we also compared our method separately with Instantstyle SDXL, which is shown in Table 2, and it can be seen that users favor our approach.

Table 1: Quantitative Experiments based on user study.

|  | Ours | InstantStyle SDXL | InstantStyle SD1.5 | DreamStyler | Inst |
|---|---|---|---|---|---|
| Style Consistency | 29.17% | 21.67% | **34.17%** | 8.33% | 6.67% |
| Face Consistency | **63.33%** | 10.83% | 6.67% | 9.17% | 10% |
| Overall | **54.17%** | 10.83% | 15.00% | 10.83% | 9.17% |

Table 2: Separately Compared with Instantstyle SDXL

|  | Ours | InstantStyle SDXL |
|---|---|---|
| Style Consistency | **52.27%** | 47.73% |
| Face Consistency | **63.64%** | 36.36% |
| Overall | **71.59%** | 28.41% |

# B    Comparative Experimental Analysis

In this section, we conduct a comparative analysis of the prompts within the AFun module discussed in Section 3.2, the two experimental implements within the CFun module detailed in Section 3.3, and the hyperparameters of the Face and Style Imagery Alignment Loss as outlined in Section 3.4.

### B.1    *Single pseudo-word Prompt* vs *Facial Description Prompt*

It can be seen in Figure 7 that the facial stylization results using facial description text prompts show better stability and generation quality, which also proves that facial description text prompts improve the accuracy of style and facial learning. Meanwhile, since facial description texts contain more information, we can obtain a better initial abstract feature, which reduces the difficulty of subsequent training. In addition, facial stylization results using only a single pseudo-word will always be affected by artifacts, especially in results with stronger styles. This further demonstrates the effectiveness of our facial description text prompts.

### B.2    Hyperparameter Analysis of Imagery Alignment Loss

We also set different $\gamma$ and $\beta$ parameters, which are 0.6:1, 0.8:1, and 1:1. It is obviously shown in Figure 7 that as a continues to increase, the style information contained in its facial stylization results will become more dense. At 0.6:1, the style information is almost completely ineffective, and its stylization ability is extremely limited, only staying at color changes or facial shifts. When set to 1:1, dense style information is introduced into the facial image, and even style leakage may occur, such as turning hair into green. This also proves the effectiveness of our proposed Imagery Alignment Loss and the existence of our proposed Imagery Latent Space. We can indeed generate stylized facial images by finding a point in the Imagery Latent Space that lies between the style image and the facial image.

Face  Style

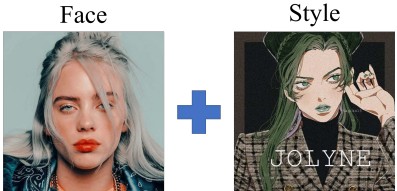

With single pseudo-word

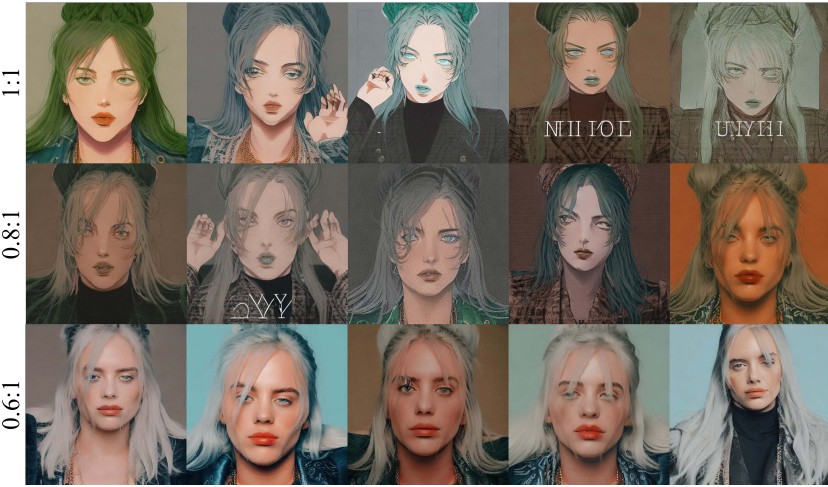

With facial description text prompt

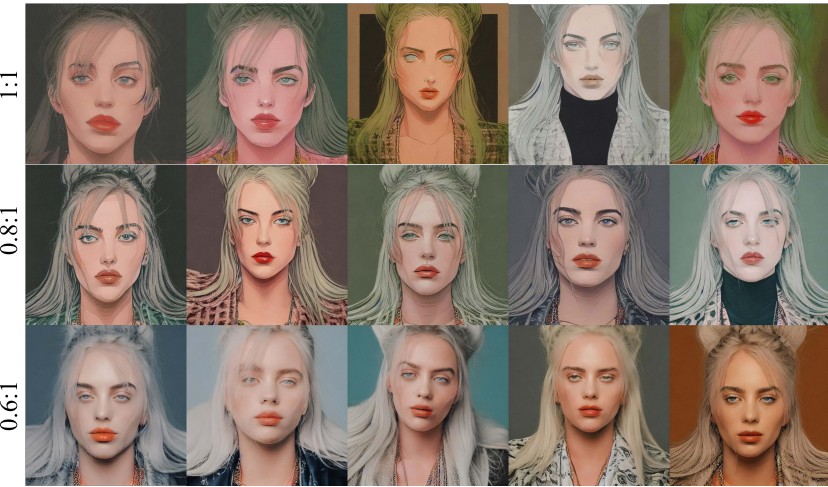

Figure 7: *Single pseudo-word Prompt* vs *Facial Description Prompt* and Hyperparameter Analysis of Imagery Alignment Loss

### B.3  *Resblock* vs *Attention Block*

We mentioned our strategy of selecting the insertion parameter position n section 3.3. The experimental results in Figure 8 indicate that inserting training parameters into attention can lead to serious style leakage, resulting in overfitting the given style image. This is because the initial abstract features $e_a$ we provide are obtained by fusing the CLIP encoding $e_{i_s}$ of the style image $I_s$ with the CLIP encoding $e_{t_f}$ of the facial description text prompt $t_f$. This means that the initial abstract features $e_a$ do not contain facial information. If we use the method of inserting parameters into attention, the style information in the abstract features $e_a$ will be activated before the information in the facial image, leading to serious style leakage and overfitting.

| Style | Face | CFun | Tuning Attention | With Mean |
|-------|------|------|------------------|-----------|

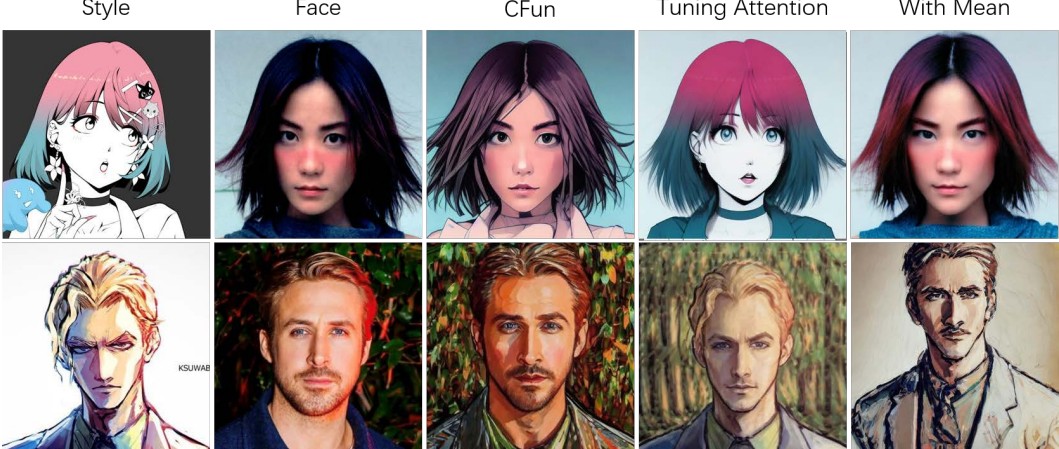

Figure 8: *Resblock* vs *Attention Block* and *Mean + Standard* vs *Only Standard*

### B.4  *Mean + Standard* vs *Only Standard*

The use of Mean becomes even more complex, as shown in the figure. We found that the reverse direction of overfitting in the final result is uncertain, possibly due to the more complex information contained in the Mean and the higher degree of mixing of facial and style information. However, regardless, the excessive content information contained in Mean is not conducive to our facial stylization. By only using Standard, we can avoid this trouble and obtain more stable facial stylization results.

## C  More Stylization Results

In this section, we conduct an interesting experiment on splicing facial images and present an array of stylized experimental results, all of which demonstrate commendable efficacy.

### C.1  Splicing Facial Stylization

We have designed an extremely interesting task to demonstrate the superiority of our proposed method and why we believe that separating abstract and concrete features for learning and finally mixing them into imagery features can effectively complete the task of facial stylization. We use this image to stylize faces by concatenating different parts of different faces into a single face. As a comparison, InstantStyle, due to the presence of ControlNet, directly generates a fragmented stylized facial image based on the cropped image, indicating that it does not understand what a face is. Our images can generate complete and meaningful stylized facial images while ensuring the features contained in each facial feature as much as possible and harmoniously integrating them into one face. This is all due to our approach of abstract, concrete, and imagery features, which makes our model adopt a more knowledge-driven approach rather than a simple feature transfer facial stylization method. This also indicates that even if the given image is fragmented, it does not prevent our method from finding a meaningful feature corresponding to it in the imagery latent space, which proves that our method has a strong abstraction ability.

### C.2  More Quality Experimental Results

We conducted experiments on more faces and styles to demonstrate the stability and generalization of our method, and we compared it with InstantStyle-SDXL, which is the best-performing InstantStyle method for comparison. in Figure 10 and Figure 11. Even though we use SD1.4, which has a smaller CLIP and training on a smaller dataset than SDXL, our method appears more adept at facial stylization tasks due to the injection of additional knowledge. It can be seen that the stylized facial images generated by our method have a high artistic level, just like the creation of an artist. Of

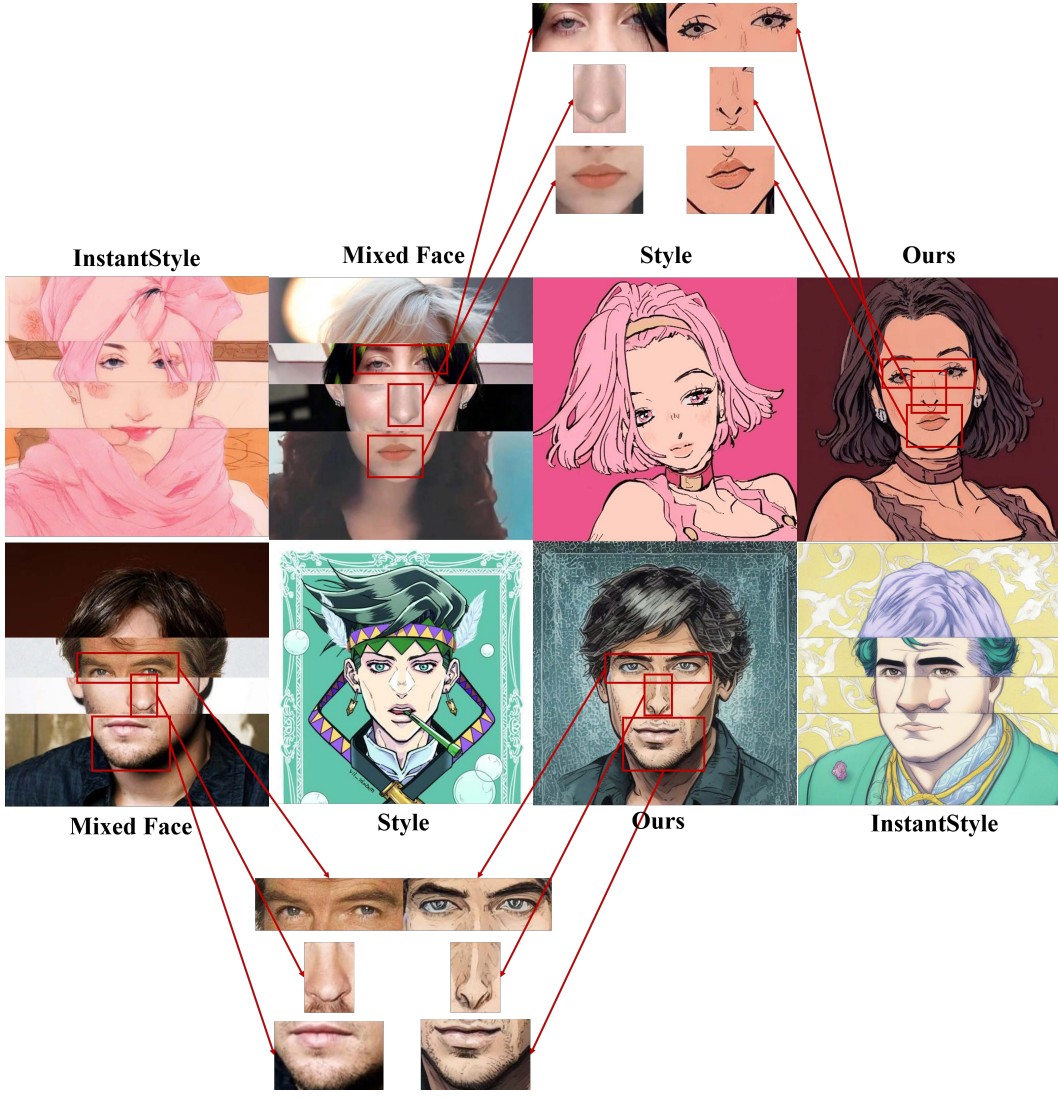

Figure 9: Splicing Facial Stylization

course, as the saying goes:' There are a thousand Hamlets in a thousand people's eyes', and people's understanding of images is subjective, triggering another reflection on me.

## D    Discussion

Is the Pre-trained Large Model a Natural Brain-like Mechanism? The fundamental idea of artificial intelligence is to simulate the operation of the human brain through artificial neurons. From a neurological perspective, human vision is not simply a peripheral sensory experience but a product of the coordination between the brain and the senses. So, the perceptual process of human "seeing" is not a simple bottom-up information transmission process but rather a top-down information construction process. The processing of facial information and style information by the human brain is unique. This also means that the action of "seeing" is not simply a visual drive but rather a knowledge-driven action.

The human visual mechanism is quite complex, especially when facing faces and artistic-style images. For faces, humans not only receive bioelectric signals from edge sensors but also activate specific brain regions and use more advanced knowledge from the hippocampus to recognize faces jointly. And the same goes for style. As a visual art experience, neuroaesthetics points out that art works

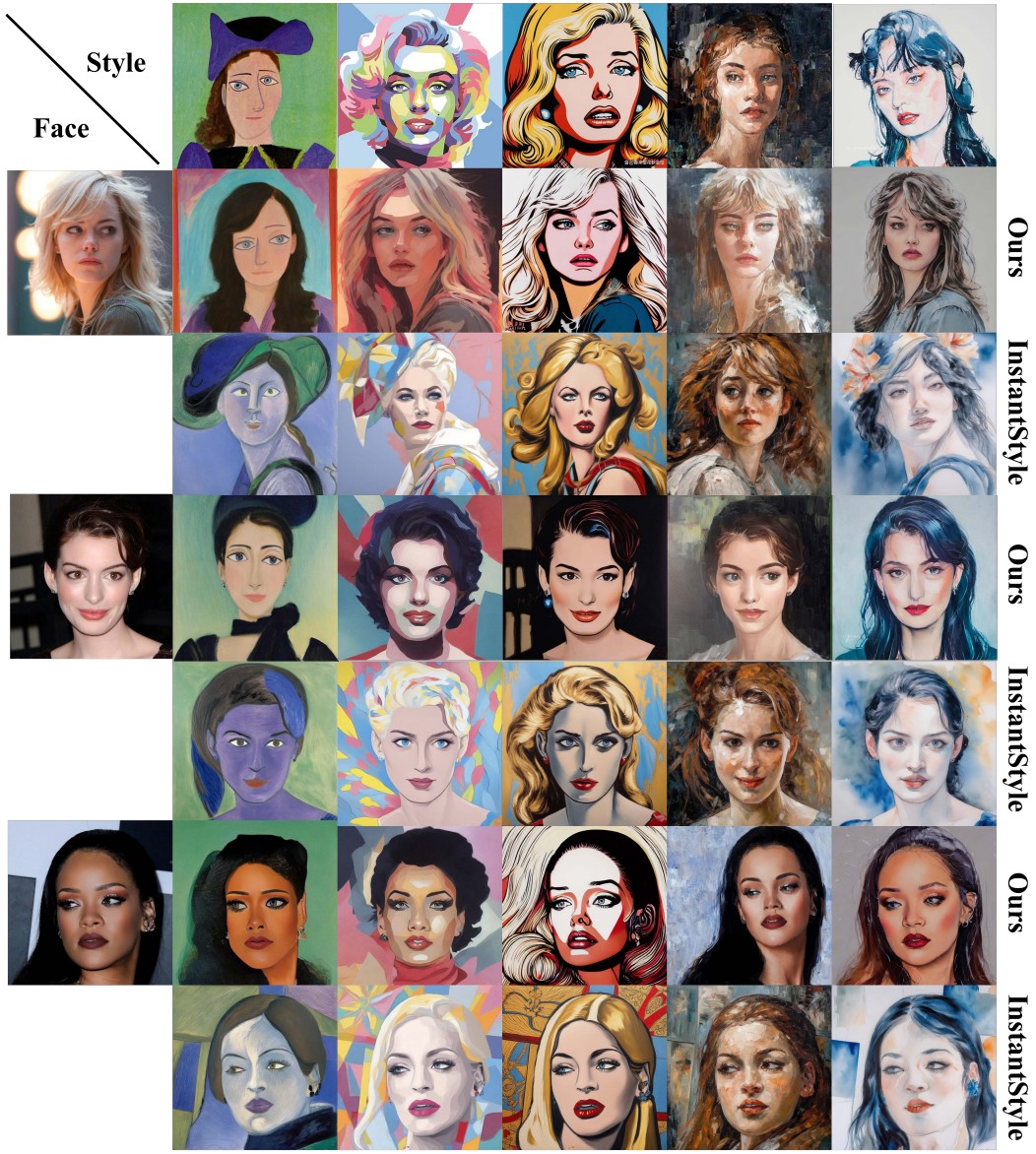

Figure 10: More Quality Experimental Results

will stimulate specific brain regions through visual stimulation, while this process still involves knowledge from the hippocampus. This is why "There are a thousand Hamlets in a thousand peoples eyes". Different people have different understandings and feelings towards the same artistic image, as it involves their past life, experiences, and educational level. Therefore, facial stylization is an extremely complex task which involves both algorithm design and human visual perception. This is why it is difficult to obtain a standard evaluation standard.

And does our method really possess mechanisms and abilities similar to the human brain? We can find a very interesting phenomenon. For the abstract and concrete features we learn, it is like the processing of visual information by the human brain. On the one hand, it is stimulated by edge sensors, and on the other hand, it is knowledge-driven cognition. In addition, concrete features also have vastly different characteristics in encoders and decoders, just like how the human left and right brains focus on different functions and tasks. However, the diffusion model is not a brain-like neural network designed concerning brain structures, which has also sparked our thinking about the relationship between large models and brain-like mechanisms.

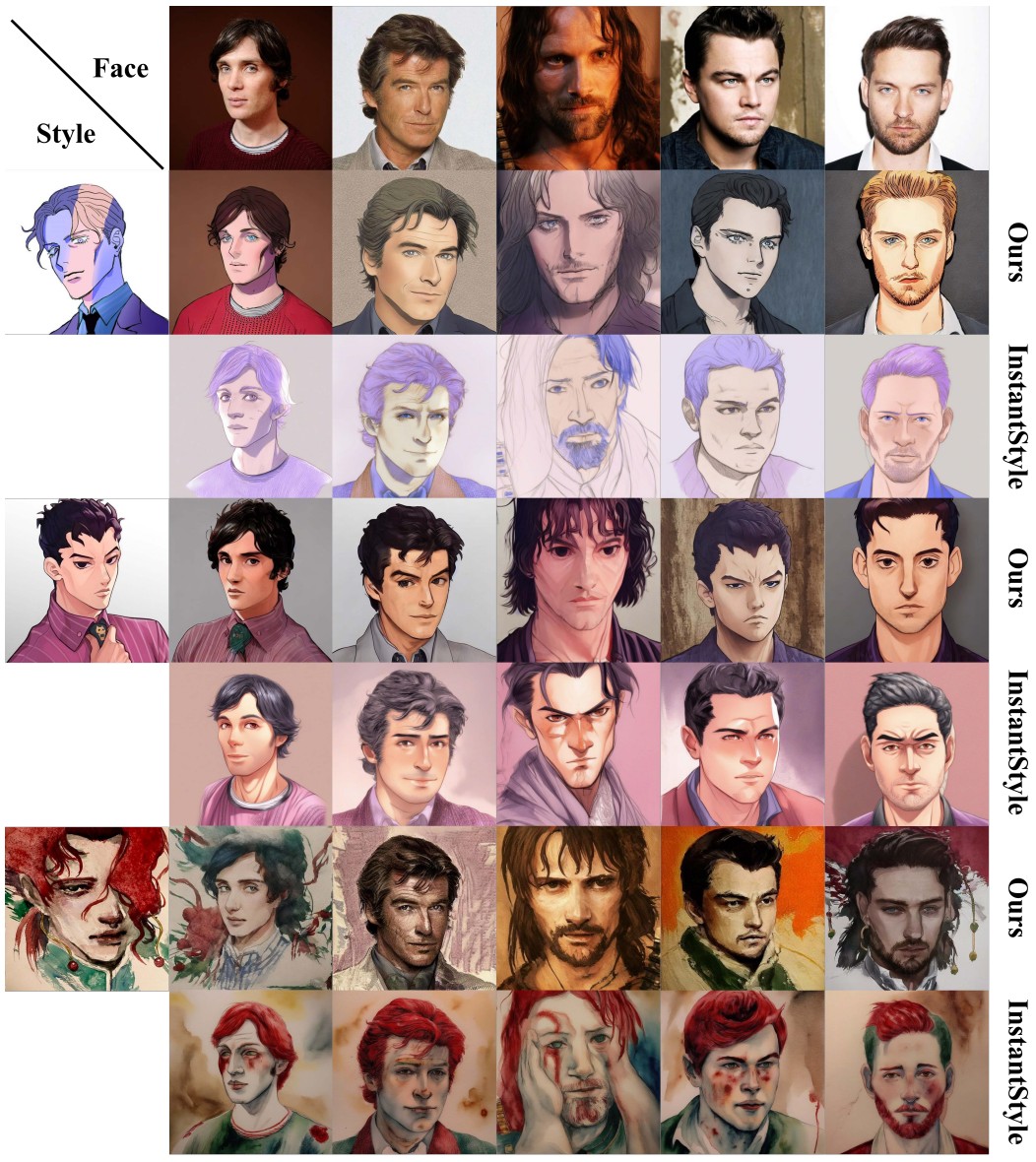

Figure 11: More Quality Experimental Results

