# OpenReview forum: "ACFun: Abstract-Concrete Fusion Facial Stylization"
_NeurIPS.cc/2024/Conference — NeurIPS 2024 poster_

### Official Review · Reviewer_uyVB · 2024-07-03

**Soundness:** 2
**Presentation:** 3
**Contribution:** 3
**Rating:** 6
**Confidence:** 3

**Summary:**

The paper introduces a new method for face stylization. The model uses a pretrained diffusion model (Stable Diffusion 1.4) and the proposed Abstract and Concrete Modules (AFun and CFun modules) to achieve the goal. The former extracts the style details to be imbued into the generative process. The latter conditions the diffusion process on those details. Notably, CFun leverages a technique known in AdaIN-like methods to modify the statistics in the generative process, while the diffusion keeps the general content of the face. Since the training happens every time a new image-style pair is introduced, it generalizes to new styles outside the used dataset. Additionally, the training lasts only 3 minutes.  This quick adaptation makes the method highly practical for users who want to efficiently apply different styles to faces.

**Strengths:**

- The goal of the paper is clearly stated and easy to understand
- Figure 2. is simple enough for a reader to grasp how the method works internally
- The proposed approach generalizes to novel face-style pairs as it is "fine-tuned" each time
- The authors present sufficient qualitative results
- I enjoyed reading the explanation behind the "abstract" and "concrete" features, which clearly show the necessity of such decoupling in the pipeline
- Splicing Facial Stylization is an exciting application of the method, which I have not seen in past works. It opens a new venue for artists.

**Weaknesses:**

- The authors tackle a highly subjective problem in terms of what makes "a good style transfer". The problem exists especially here, as only qualitative results are provided. For example, in many cases, InstantStyle SDXL provides (in my subjective opinion) better results than the proposed method. Additionally, Tuning Attention exhibits a better style transfer than CFun, as shown in Figure 8 in the supplementary. Figure 11 confuses me even more as the proposed method does not transfer the style correctly compared to InstantStyle.
- Although the introduction clearly states the problem and how the problem is being solved, the latter sections contain vague sentences that make the overall method hard to comprehend. For example, I do not understand how the alignment loss (Section 3.4) aligns the images as the diffusion process is tasked to reproduce both the style and reference images using the same input. The section does not say why the alignment is needed, and the ablation study does not explain it either. The right section of Figure 3. (especially the right bottom image) makes it even more ambiguous.
- Some parts need further explanation, for example:
    - (L65) What does making an image recognizable by the CLIP model exactly mean?
    - (L51-L53) The sentence is hard to understand what needs balancing
    - (L193) What is VQ space?
    - (L234-L237) What are those traditional methods, and what is the "two-column method"?
    - (L241-L243) What do the authors mean by "understanding the face" by a model? If ControlNet can reconstruct a face, does it not mean it understands the face, too?
    - (L286-L287) What is the binding phenomenon, and what does the style leakage look like?

**Questions:**

Questions:
- Why did the authors not apply "Tuning Attention" as the main component of the method instead of CFun? It seems to transfer the style more reliably and serves the same purpose as CFun?
- Could the authors elaborate on how using this method makes the stylization less effortless than using InstaStyle SDXL?

Suggestions:
- The authors need to evaluate the model in terms of the user preference. Given a high enough number of participants, the survey would objectively tell which stylization is better for an average person.
- I suggest clarifying the parts mentioned in the weaknesses, which I attach below for self-consistency
- CFun resembles the commonly used AdaIN module
- Some parts need additional references. For example, the diffusion described in L228 seems to be a DDIM [2]

[1] Huang X, Belongie S. Arbitrary style transfer in real-time with adaptive instance normalization. In Proceedings of the IEEE international conference on computer vision 2017 (pp. 1501-1510).
[2]  Jiaming Song, Chenlin Meng, Stefano ErmonDenoising Diffusion Implicit Models. In International Conference on Learning Representations 2021.

**Limitations:**

- The authors describe the limitations in a separate section. However, some additional visual examples would be necessary to explain what the authors mean by "extract a few visual elements of the style we want to exist in the style image" (L283) and "style leakage" in 287.

---

> ### Author Rebuttal · Authors · 2024-08-07
>
> Thank you for recognizing our work and pointing out our issues in detail. Thank you for taking the time to review our work. We would like to provide the following answers to the questions you have pointed out.
> W1: You have pointed out a crucial issue, and we have added quantitative experiments based on user studies. It should be noted that our focus is not simply on style conversion tasks but on stylizing faces. This requires us to maintain various information in the original face image, such as identity, expressions, etc. while stylizing. However, it can be seen that although InstantStyle has a strong style, its ability to maintain facial information is insufficient, especially in Figure 11, where we can see that it cannot faithfully restore the identity information of the face, especially in cases 1 and 3. In addition, similar to the above requirements, in order to ensure facial information as much as possible, we choose not to tune the attention, as shown in Figure 8. While tuning the attention can achieve a strong style transfer effect, it also masks the content contained in the original facial image, making it difficult to distinguish the facial identity information of the original image. This is not the effect pursued in our article. So, we adopt the CFun method for style transfer to reduce the impact of content information in the style image on the original image.
>
> W2: The alignment loss we propose is designed for the training paradigm we use. As you mentioned, general diffusion models use a single image input and output, followed by constraints. However, we use the method of simultaneously inputting two images and finding a balance point between the style image and the face image in the latent space to complete the face stylization task. Through this approach, our method has the ability to migrate higher-level visual semantic elements. In Supplementary Material Figure 7, the results of training with different proportions of alignment loss in latent space are presented, demonstrating the effectiveness of our alignment loss and demonstrating how this latent space equilibrium point works. The image on the right of Figure 3 reflects the latent space we mentioned earlier. Its specific meaning lies in that the vq space used in stable diffusion is only a low-level visual element, and directly searching for balance points in vq space is often meaningless. Instead, we use a combination of vq embedding and clip attention to search in the latent space to obtain a meaningful balance point between the style image and the face image, thus achieving the task of facial stylization
>
> W3:
> 1. Due to the presence of hidden words * in the input text, such clip embeddings usually have no practical significance for subsequent UNETs. Therefore, we hope to optimize this clip embedding through training so that subsequent UNETs can correspond to this clip embedding with the balance point we previously proposed. This process is called "making an image recognizable by the CLIP model"
> 2. As mentioned earlier, we are searching for a balance point between the style image and the face image in the latent space, which not only contains the style information we need but also faithfully preserves the content information in the face image
> 3. The stable diffusion we use owns the VQ (vector quantization) space, which encodes points in a vector space using a finite subset of points to complete image compression and accelerate the speed of image generation by the diffusion model.
> 4. The two columns on the right are the traditional methods. These methods only transfer low-level visual features such as color and texture and do not involve high-level semantics such as shape. While all other methods can change high-level semantics.
> 5. The method used by Instantstyle is to input edge information into ControlNet to control the final generation result. We believe that this approach does not represent an understanding of the face. From its results, it can also be seen that it does not have the ability to maintain facial identity information. In addition, as shown in Fig. 9 of our supplementary materials, InstantStyle cannot achieve facial stylization by Splicing face images using ControlNet. At the same time, our method can integrate them into a complete face, which also demonstrates our claimed ability to understand faces.
> 6. The binding phenomenon refers to the strong correlation between some content information and style information, which leads to the appearance of some content in style images as a result of facial stylization, which is also called style leakage. A typical example is our result in case 3 of Figure 4. In case 4, we can see that the hand in the style image appears in case 3, and the color of Leonardo's suit in case 4 is consistent with the given style image.
> Q1: We hope to preserve the content information in the facial image as much as possible. Although tuning the attention can result in a stronger style, as shown in Figure 8, this generated result changes the facial image beyond recognition, making it difficult to identify its identity information. We also need to avoid this situation as much as possible in our work so we do not tune the attention.
> Q2: Firstly, it should be noted that InstantStyle uses 4 million image text pairs to train the IP-Adapter used, while we only use one pair of face images and style images, which demonstrates the advantages of our method in terms of computational resources and cost. Secondly, as a facial stylization task, we faithfully restored the identity information of the given person's facial image, allowing the stylized result to still be recognized as identity. However, InstantStyle performed poorly in this regard.
>
> S1: We have supplemented quantitative experiments based on user studies
> S2: shown above
> S3: We referred to Adain to some extent when designing the Cfun
> S4: Thank you for pointing it out. We will add these references

---

> > ### Comment · Reviewer_uyVB · 2024-08-11
> > **Response to the Authors' Rebuttal**
> >
> > Thank you for your detailed feedback and for providing the results in PDF format. I appreciate that most of your concerns were addressed in the response.
> >
> > The primary issue affecting my initial score was the absence of quantitative results. Now that this has been rectified, I am pleased to increase my final rating. Additionally, I appreciate the design of the study and the decision to divide the evaluation into three independent aspects.
> >
> > I have one remaining question (which, however, does not impact my score): Could you specify which components are pretrained? For instance, in Stable Diffusion, the VQGAN component is pretrained prior to training the diffusion model. Including such details in the implementation section would enhance the method's comprehensiveness.
> >
> > My main remaining concern is the quality of the writing. There are several instances where the flow is disrupted or where there is repetitive use of words, which could potentially confuse the reader. For example:
> > - Line 133: The introduction of the problem with inversion-based models over two sentences causes the reader to lose track of the method's overview.
> > - Line 187: The sentence is overly long.
> > - Lines 197-201: These lines are difficult to comprehend on first reading.
> >
> > Addressing these issues could significantly improve the clarity and readability of the text.

---

> > > ### Author Response · Authors · 2024-08-11
> > >
> > > Thanks for taking the time to provide such a detailed response and acknowledging our work.
> > >
> > > For the stable diffusion model and VQGAN question you mentioned, the vq encoder is a module in the pre-trained stable diffusion model used to accelerate it. The stable diffusion training process first trains the VQ encoder and then the diffusion model itself.
> > >
> > > Also, thank you so much for your suggestions on our writing. We will follow your suggestions to improve the fluency of our paper, avoid long and difficult sentences as much as possible, and use clearer expressions to improve our writing.

---

### Official Review · Reviewer_6WJE · 2024-07-04

**Soundness:** 3
**Presentation:** 3
**Contribution:** 3
**Rating:** 7
**Confidence:** 4

**Summary:**

This paper proposes a novel facial stylization method called ACFun, which achieves high-quality stylization effects by combining abstract and concrete visual elements. It contains the Abstract Fusion Module (AFun) and Concrete Fusion Module (CFun) to learn the abstract and concrete features. A new loss function is designed to align style and facial images in latent space to improve stylization accuracy. The effectiveness of the ACFun method is validated through extensive experiments, which can produce more artistic and visually appealing facial stylization results compared to existing methods.

**Strengths:**

1)	A novel facial stylization method called ACFun is proposed, which solves the limitations of existing stylization methods in processing facial images by combining abstract and concrete features.
2)	Through extensive experiments, the author demonstrated the effectiveness of the ACFun method in facial stylization tasks, which can produce higher quality artistic results compared to other existing technologies.
3)	The paper has a clear structure and rigorous logic, with clear organization from problem introduction to methodology, experimental results, and discussion, making it easy for readers to understand and follow up.

**Weaknesses:**

Some experimental results still need to be supplemented to demonstrate the generalization ability of this method：
1)	Will there be any difference in the final results generated using detailed and vague text descriptions for the facial description prompts proposed in this paper?
2)	The experiment only showed the generation results of 40 diffusion steps. What are the differences between different diffusion steps?
3)	In the experiment, the gender of the given style image and face image is always the same. Does this method still work when faced with the gender of the given style image and the face image is different?

**Questions:**

Shown as weaknesses.

**Limitations:**

Shown as weaknesses.

---

> ### Author Rebuttal · Authors · 2024-08-07
>
> Thank you for recognizing our work and pointing out our issues. We have supplemented the experiment you pointed out in Weakness. It can be seen that fewer steps will make the image more realistic, while more steps will make it more stylized. For different text descriptions, it can be seen that we have adopted more detailed facial descriptions and more detailed style image descriptions, which will also affect the bias of the final generated results. Finally, it can be seen that our method can still work even in different gender situations, which proves our method's decoupling ability and stylization accuracy.

---

> > ### Comment · Reviewer_6WJE · 2024-08-14
> >
> > Thanks for the author's response. I will keep my original score.

---

### Official Review · Reviewer_5hdY · 2024-07-11

**Soundness:** 3
**Presentation:** 3
**Contribution:** 2
**Rating:** 4
**Confidence:** 4

**Summary:**

This paper deals with the problem of facial stylization using one style image and one facial image. Specifically, the authors design an abstract fusion module and a concrete fusion module to learn the abstract and concrete features of the style and face separately. They further design a face and style imagery alignment loss for face and style image aligning. Experiments show that the proposed method outperforms other methods.

**Strengths:**

1. The topic is interesting, nowadays, face stylization using diffusion models is getting more and more attention.
2. The paper is easy to follow and the proposed method is intuitive and easy to understand.
3. Many visualization results are shown in well-plotted figures.

**Weaknesses:**

1. There are no quantitive metrics, which can show the proposed method is better than the existing methods, only qualitative ones, which might be subjective and biased.
2. Some visualization results still have room to be improved. The most concern is that some style information is merged with the identity or object information of the style image, which is not well disentangled. For example, in the Fig.11, the first style image have purple and pink style, while the proposed method cannot learn the detailed style information very well.
3. The method is lack of novelty. To me, using the guidance of CLIP image and text features is somehow outdated. The concrete fusion module is an attention module to me, which is also well studied.

**Questions:**

Could the authors provide more information about using CLIP, as it seems to me there are better vision language models which can extract better features than CLIP? How about using other models?

**Limitations:**

The authors have discussed the limitations of their paper.

---

> ### Author Rebuttal · Authors · 2024-08-07
>
> Thank you for admiring our work and pointing out our issues. We respond to the weaknesses and questions you pointed out as follows.
> W1: We quantify style, content, and overall through user study. We selected 50 people through a survey questionnaire, and the results showed that our method achieved better results
> W2: Our method aims to reproduce the content of the reference image as much as possible rather than the style image. The results here demonstrate that our method successfully separates "purple and pink style" from its style and only transfers the expression of the style image to the target image.
> W3: We adopted CLIP because our backbone is Stable Diffusion, and the language model used is CLIP. Therefore, we also adopted CLIP.
> Q1: Due to the use of CLIP in the backbone, DINO is difficult to adapt.

---

> > ### Author Response · Authors · 2024-08-13
> >
> > I'm sorry, the answer to Weaknesses 3 and the question were ambiguous, which may cause confusion for you. I want to add that due to the use of backbone stable diffusion as a text-to-image generation model, the language model used is CLIP. As a pre-trained large model, it is necessary to retrain the diffusion model to replace CLIP with another model. However, the training cost of diffusion models is generally difficult to afford, so we choose to continue using CLIP in stable diffusion rather than choosing other new visual language models. In addition, the reason why we use CLIP as a visual extractor is also because the image embeddings encoded by CLIP's image encoder are more easily aligned with the text embeddings in the latent space they construct, which is also beneficial for subsequent fusion by cross-attention.

---

### Official Review · Reviewer_bo1j · 2024-07-17

**Soundness:** 1
**Presentation:** 1
**Contribution:** 2
**Rating:** 4
**Confidence:** 5

**Summary:**

This article introduces a generative model ACFun for facial stylization, which designs two modules AFun and CFun to learn the abstract and concrete features of styles and faces. The authors design a Face and Style Imagery Alignment Loss to align the style image with the face image in the latent space, using these methods to extract more levels of style features and better balance facial changes caused by stylization and preserving facial information.

**Strengths:**

1. This article achieves style extraction at both abstract and concrete levels with only one pair of images.
2. The Face and Style Imagery Alignment Loss proposed in this article is interesting as it seeks the balance point between style images and facial images in latent space.

**Weaknesses:**

1. Lack of Quality Analysis. Sec. 4.1 is missing.
2. The ability of text-guided image generation was introduced in the experiment but only compared with InstantStyle, which seems insufficient.
3. The role of CFun in the encoder and decoder was analyzed in the ablation experiment, and the results are shown in Fig.6. However, for Encoder, the result in the first line is only a blurry image with a rough structure, while the second line shows the specific facial structure and posture. It seems that the conclusions drawn from the two examples are different. For the decoder, the first line shows the specific facial structure and posture, while the second line's facial structure and posture appear to be inconsistent with the reference image. These two examples are insufficient to prove that “In the decoder, CFun can see that the main concrete features are the structure and posture of the face.”

**Questions:**

1. What is y in Equation 1?
2. The reviewer is interested in the image on the right side of Figure 3 and suggests adding some visualizations or experiments to demonstrate the role of the loss function in aligning style and face images.

**Limitations:**

Limitations are discussed.

---

> ### Author Rebuttal · Authors · 2024-08-07
>
> Thank you for pointing out the problems in our paper. We respond to the weaknesses and questions you pointed out as follows.
> W1: I'm very sorry. Due to a layout problem, the content of section 4.1 is in the second paragraph of section 4.2.
> W2: Due to the excellent performance of InstantStyle, we only compared the results of this method. Now, we have added a comparison of the results between Dreamstyler and Inst.
> W3: We have added more visual results of ablation experiments. For abstract features, there are indeed certain patterns in the encoder and decoder for specific features, manifested as a bias towards color, texture, and structure, facial features, respectively.
> Q1: This method takes two images as input, where x represents the face image, and y represents the style image
> Q2: In the supplementary materials, Figure 7 demonstrated the effect of alignment loss in latent space, and we also conducted additional experiments

---

> ### Comment · Reviewer_bo1j · 2024-08-13
>
> Thanks for the response. However, my concerns for not suggesting acceptance were not solved:
> First, the method itself is not groundbreaking. The overall pipeline is similar to many popular solutions such as IPadaptor, which extract image and text features for a few shot samples and then fuses them to the pretrained network. The proposed CFun module is also rather simple without well-explained motivation.
> Second, the results are not impressive. The authors claim that they only compared with InstantStyle for its excellent performance. However, InstantStyle is not even a peer-reviewed technical paper. Such a claim made the reviewer challenge the authors' basic technical skills. The newly added statistics are also not convincing. The face consistency of the proposed method is extremely high then the other method. However,  the visual results could not support such statistics. Also, the stylized effects are worse than the other selected methods.

---

> > ### Author Response · Authors · 2024-08-13
> >
> > Thanks for your response. Firstly, unlike IP adapters, which aim to fuse a concept into a diffusion network using few shot images, our approach seeks to merge style images and facial images into one concept. In addition, we conducted comparative experiments comparing the results of the peer-reviewed papers Dreamstylist and Inst. We explained in the paper that we divide image features into abstract and concrete visual features for subsequent face stylization tasks. CFun was designed with this motivation and demonstrated its effectiveness, it is designed to transmit low-level visual features to make stylized faces more similar in specific strokes, textures, and other aspects. And simplicity itself is not a disadvantage. Simple and effective designs are often easier to understand and implement. Finally, regarding your subjective judgment on our method, as reviewer uyVB pointed out, this is a highly subjective issue, and a few individuals alone may have divergent judgments on the final result. That's why we cited a user study to demonstrate the superior performance of our method's results. Finally, up to now, the IP adapter you mentioned, like InstantStyle, is also not a  peer-reviewed paper, and even if InstantStyle is not a peer-reviewed paper, if it performs well in practice, it can still serve as an effective benchmark for comparison. In addition, we will add more visual results in the supplementary materials in future versions.

---

> > > ### Author Response · Authors · 2024-08-13
> > >
> > > Regarding novelty:
> > > Reviewer 6WJE proposed: A novel facial stylization method called ACFun is proposed, which solves the limitations of existing stylization methods in processing facial images by combining abstract and concrete features.
> > > Reviewer uyVB proposed: The proposed approach generalizes to novel face-style pairs as it is "fine-tuned" each time. Splicing Facial Stylization is an exciting application of the method, which I have not seen in past works. It opens a new venue for artists.
> > >
> > > Regarding the motivation of CFun design:
> > > Review uyVB proposed: I enjoyed reading the explanation behind the "abstract" and "concrete" features, which clearly show the necessity of such decoupling in the pipeline.
> > > Reviewer 6WJE proposed: A novel facial stylization method called ACFun is proposed, which solves the limitations of existing stylization methods in processing facial images by combining abstract and concrete features.
> > >
> > > Regarding the experiments:
> > > Reviewer 5hdY proposed: Experiments show that the proposed method outperforms other methods. Many visualization results are shown in well-plotted figures.
> > > Reviewer 6WJE proposed: Through extensive experiments, the author demonstrated the effectiveness of the ACFun method in facial stylization tasks, which can produce higher quality artistic results compared to other existing technologies.
> > > Reviewer uyVB proposed: The authors present sufficient qualitative results.

---

### Author Rebuttal · Authors · 2024-08-07

We have added experiments on text-guided generation and ablation and related experiments on different diffusion steps, different levels of text description detail, and different genders. We also conducted quantitative experiments based on user study, providing 40 pairs of style facial images and their generated results, and collected voting results from 50 volunteers. The voting was conducted from three aspects: style consistency, facial consistency, and overall which image users prefer. The results are shown in the table. It can be seen that our method can achieve good results in style consistency and outstanding performance in facial consistency and overall evaluation. In addition, we also compared our method separately with Instantstyle SDXL, and it can be seen that users favor our approach.

---

### Decision · Program_Chairs · 2024-09-25

**Decision:**

Accept (poster)

**Comment:**

This paper received mixed reviews, ranging from "Borderline reject" to "Accept". The reviewers noted that the paper's novel facial stylization, ACFun, achieves style extraction at both abstract and concrete levels using only one pair of images, calling it well-structured and easy to follow, and acknowledged its extensive experiments and visualizations.

Many of the reviewer's concerns related to the paper's experimental setup: The reviewers noted a lack of quantitative metrics and potential subjectivity in qualitative results, though this was partially addressed in the rebuttal  (5hdY, uyVB). The reviewers also called out limited comparison with existing methods, particularly for text-guided image generation (bo1j, 5hdY), inconsistent / unclear experimental results, particularly in ablation studies and style transfer quality (bo1j, uyVB), and the need for additional experiments to demonstrate generalization ability and address specific scenarios (6WJE). Other concerns related to insufficient novelty (5hdY) and unclear technical details (uyVB).

The authors provided a detailed rebuttal, but it, and the reviewer discussion, did not change the distribution of recommendations. In such borderline cases, I am inclined to err on the side of acceptance and let the community decide the merits of this work.